# Asymmetry and non-dispersivity in the Aharonov-Bohm effect

Maria Becker[1], Giulio Guzzinati [2], Armand Béché [2], Johan Verbeeck[2] & Herman Batelaan[3]

Decades ago, Aharonov and Bohm showed that electrons are affected by electromagnetic potentials in the absence of forces due to fields. Zeilinger's theorem describes this absence of classical force in quantum terms as the "dispersionless" nature of the Aharonov-Bohm effect. Shelankov predicted the presence of a quantum "force" for the same Aharonov-Bohm physical system as elucidated by Berry. Here, we report an experiment designed to test Shelankov's prediction and we provide a theoretical analysis that is intended to elucidate the relation between Shelankov's prediction and Zeilinger's theorem. The experiment consists of the Aharonov-Bohm physical system; free electrons pass a magnetized nanorod and far-field electron diffraction is observed. The diffraction pattern is asymmetric confirming one of Shelankov's predictions and giving indirect experimental evidence for the presence of a quantum "force". Our theoretical analysis shows that Zeilinger's theorem and Shelankov's result are both special cases of one theorem.

[1] Department of Physics, Hastings College—Morrison-Reeves Science Center, Hastings, NE 68901, USA. [2] EMAT, University of Antwerp, Groenenborgerlaan 171, 2020 Antwerp, Belgium. [3] Department of Physics and Astronomy, University of Nebraska-Lincoln, 208 Jorgensen Hall, Lincoln, NE 68588-0299, USA. Correspondence and requests for materials should be addressed to H.B. (email: hbatelaan2@unl.edu)

The Aharonov-Bohm (AB) effect[1–6] entails the presence of a phase shift caused by a magnetic flux enclosed by an electron interferometer. It is thought to demonstrate the physical reality of potentials[2,3], as opposed to the earlier interpretation that potentials were merely a mathematical tool[3,4]. The reason for this change in understanding came about because the fields outside a magnetic flux tube, such as provided by a perfect solenoid (infinitely long and infinite winding density), are zero, thus eliminating the possibility of a classical Lorentz force. Under the assumption that the solenoid is unperturbed[5,6], there is no field that can act locally on the electrons. However, the non-zero vector potential can have a local effect that results in a phase shift. Notwithstanding the general acceptance of these ideas, this issue remains a topic of debate on non-locality[7–10], the interpretation[11–14], and existence of the effect[15,16].

The AB effect has been observed for free electrons in a series of ever more refined experiments[17–21], as well as in conductors[22–25]. The absence of a longitudinal force, as made apparent by the absence of electron time delays, has been investigated more recently. These time delays, predicted by alternative theories[5], have been ruled out[26]. However, deflection, another indicator of force, has been predicted by Shelankov[27], elucidated by Berry[28], and theoretically confirmed by Keating and Robbins[29]. The deflection is accompanied by a characteristic asymmetry in the electron diffraction pattern providing an experimental signature.

The presence of force has been operationally defined by Zeilinger using the expectation value of position[30]. If the expectation value differs from the value obtained for free propagation, then a force is present. For experiments with electron beams, the presence of a longitudinal force along the beam would lead to time delays in the expectation value of the arrival time, while a transverse force would lead to deflections. Zeilinger's theorem, as expounded by Peshkin[31], indicates that a characteristic feature of the AB effect is its dispersionless (i.e., force-free) nature. Experimental demonstrations of the dispersionless nature of AB-duals[32], including the He-McKellar-Wilkens effect, have been performed[33], while a demonstration of the dispersionless nature of the magnetic AB effect has yet to be reported[34,35].

In this paper, we report the observation of electron diffraction asymmetry consistent with theory. To this end, an electron beam is passed through a small aperture that holds a magnetized nanorod. It is confirmed that reversal of the magnetization direction reverses the observed asymmetry. This experimental result provides support for Shelankov's theoretical prediction, and thus indirectly indicates the presence of force. A crucial experiment remains necessary to directly demonstrate the electron beam deflection by measuring the expectation value. The presence of force is in apparent contradiction to textbook descriptions of the AB effect. We report a theorem that resolves this issue by showing the absence of classical forces and the presence of quantum "forces". The absence of classical forces in the longitudinal direction of the electron's motion is consistent with Zeilinger's theorem and supported by experiment[26]. However, Zeilinger's theorem cannot be applied to the transverse motion for the AB physical system. The theorem is correct, but its assumptions are not generally applicable to the physical situation considered. Shelankov's prediction pertains to the transverse motion of the electron for the AB physical system and is supported by our experimental results. Our deflection theorem is a generalization of Peshkin's approach, and when applied to the infinitesimal flux-line yields Shelankov's and Zeilinger's results as two limiting cases. Additionally, the deflection theorem is applied to a finite-size flux tube to provide the connection to experiment.

## Results

**SB approach.** To explain the theoretical prediction, consider a coherent electron beam passing by a current-carrying solenoid as illustrated in Fig. 1. The solenoid is assumed to be ideal, i.e., it carries no stray fields and its field is not affected by the passing beam. We are interested in obtaining the far-field electron diffraction pattern. Specifically, the expectation value of the transverse position of the electron is used to assess whether or not a force acted during the passage of the electron by the solenoid. This determination of force is studied in several steps. In the first step, Berry's derivation[28] of Shelankov's result[27] is summarized. In the second step, we derive a theorem that yields Zeilinger's theorem[31] and Shelankov's result as special limiting cases. In the third step, Shelankov's result is used as a benchmark for a path integral simulation, which allows for the simulation of detailed experimental parameters. A de Broglie-Bohm viewpoint of the physical scenario is provided in step four, which serves to illustrate the term quantum "force", as introduced by Berry, and Keating and Robbins.

Berry identifies the problem as two-dimensional and describes the incoming electron wave with a superposition of multiple plane waves[28]. The incoming waves have a Gaussian distribution of wave vector directions in the $x$–$y$ plane, which yields Shelankov's result in the paraxial approximation,

$$c_{\text{paraxial}}(\alpha, \theta) = \exp\left(-\frac{1}{2}\theta^2 w^2\right) \times \left[\cos(\pi\alpha) + \sin(\pi\alpha)\, erfi\left(\frac{w\theta}{\sqrt{2}}\right)\right]$$

(1)

Here, $c(\alpha, \theta)$ is the probability amplitude for electrons to be

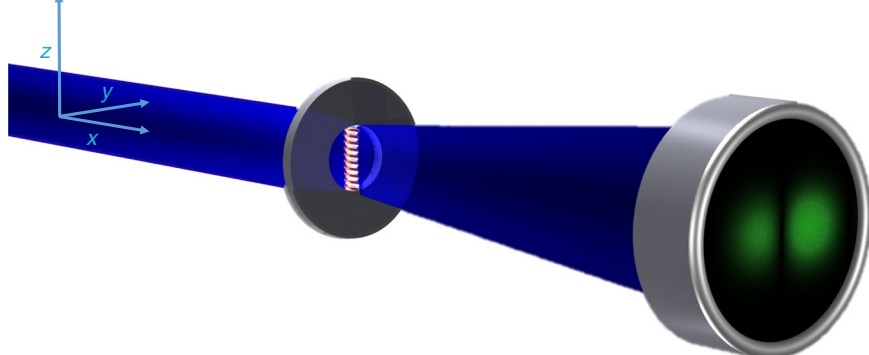

**Fig. 1** Physical system schematic. An electron beam (blue) diffracts from an aperture that holds a magnetic flux line, here represented by a solenoid. The solenoid is opaque to the electrons, and the electrons pass through an area where there is no magnetic or electric field, and thus no classical force. The non-zero expectation value of position, represented by a left-right asymmetry in the strength of the detected electrons (green), indicates the presence of a quantum "force" for the Aharonov-Bohm physical system

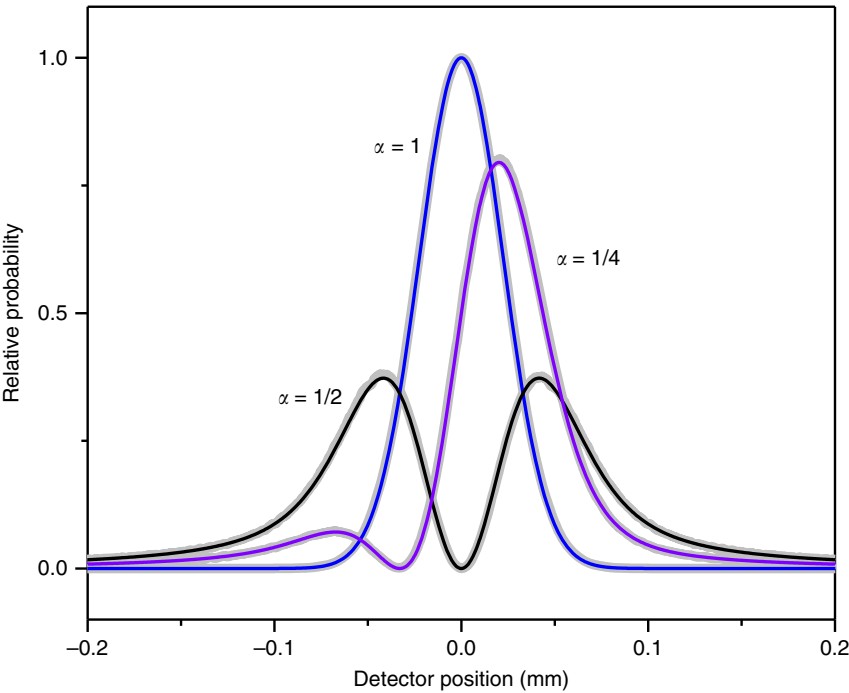

**Fig. 2** Far field electron diffraction. An electron diffraction pattern for a 1-D aperture that holds a magnetic flux line is given in the far-field. The result of a path integral simulation (thick gray lines) is in agreement with Berry's analytic result (black, blue and purple line) and is shown for three magnetic flux line of strengths, $\alpha$. A non-zero expectation value of position for the case that $\alpha = 1/4$ indicates the presence of a force for the AB physical system. The path integral simulation is developed for the purpose of including a 2-D circular aperture, a partially coherent electron beam, and a finite-sized magnetic flux bar (instead of a flux line) to facilitate a detailed comparison with experiment

scattered in the $\theta$ direction (defined with respect to the $x$-axis in the $x$–$y$ plane) for a magnetization flux, $\Phi$, of the infinitesimal solenoid (or magnetic flux line), where $\Phi$ is indicated in quantum units by $\alpha = -e\Phi/h$. The r.m.s angular width of the incident electron distribution is $1/w\sqrt{2}$. The relative probability distribution obtained from $|c(\alpha, \theta)|^2$ is shown (Fig. 2) for three different values of $\alpha$. When $\alpha = 1/4$, an assymetric probability distribution with a non-zero deflection is found. Another approach is the use of a quantum "force" operator as shown by Keating and Robbins[29]. They successfully ensure the Hermiticity of the operator and obtain the same deflection.

**Deflection theorem**. Consider an initial state ($t = 0$) of a Gaussian wavepacket in the momentum representation with a normalized momentum distribution, a width $1/a$, and a linear phase ramp proportional to $x_0$,

$$\varphi(k, 0) = \left(\frac{a^2}{\pi}\right)^{1/4} e^{-(k-k_0)^2 a^2/2} e^{-i(k-k_0)x_0}. \quad (2)$$

We assume that after an interaction the wavepacket is modified to

$$\varphi_A(k, 0) = \left(\frac{a^2}{\pi}\right)^{1/4} e^{-(k-k_0)^2 a^2/2} e^{-i(k-k_0)x_0} F(k), \quad (3)$$

where $F(k)$ is an arbitrary complex function dependent on momentum; $\varphi_A(k, 0)$ is normalized. This means that the interaction is assumed to be approximately instantaneous (which holds for the physical system studied, see Methods). After the interaction, the time-dependent wavefunction is written in the position representation as the wavepacket

$$\psi(x_i, t) = \frac{1}{\sqrt{2\pi}} \int \varphi_A(k, 0) e^{-i(kx_i - \omega(k)t)} \mathrm{d}k, \quad (4)$$

where $x_i$ is the position, which can be taken to be the transverse, $x_T$, or longitudinal, $x_L$, coordinate and $\omega(k) = \hbar k^2/2m$. The expectation value of the position operator $x_i = i\partial/\partial k$ for the wavefunction in Eq. (4) is given by the resulting deflection theorem

$$\langle x_i \rangle = x_{i0} + \frac{\hbar \langle k \rangle}{m} t + \frac{a}{\sqrt{\pi}} \int \frac{\partial \delta}{\partial k} |R|^2 e^{-(k-k_0)^2 a^2} \mathrm{d}k. \quad (5)$$

where $F(k) = R(k)e^{i\delta(k)}$ in polar coordinates. See Methods section for additional derivation steps.

**Dispersionless and quantum "forces"**. Now, we can investigate two specific cases of the interaction: (1) the Zeilinger-Peshkin (ZP) scenario, and (2) the Shelankov-Berry (SB) scenario. In the first case, it is assumed that the interaction results in a pure phase shift (see Peshkin's clear derivation[31]);

$$\delta(k) = \delta(k_L); \quad R(k_L) = 1, \quad (6)$$

where the longitudinal momentum, $k_L$, has a Gaussian distribution $e^{-(k_L - k_{L0})^2 a^2/2}$ (Fig. 3). In this case, the expectation value for the position follows directly from Eq. (5),

$$\langle x_L \rangle = x_0 + \frac{\hbar k_0}{m} t + \frac{a}{\sqrt{\pi}} \int \frac{\partial \delta}{\partial k} e^{-(k-k_0)^2 a^2} \mathrm{d}k. \quad (7)$$

This is Zeilinger's dispersivity theorem[31]. In words, it states that when an interaction is dispersionless (i.e., $\partial\delta/\partial k = 0$), there is no shift of the wavepacket's position expectation value compared to its classical counterpart. It has motivated experiments that demonstrate the dispersionless nature of the AB-effect[32,33], which are interpreted to mean that the AB-effect is force-free.

In the second case, the interaction is assumed to lead to a phase step in position, $F(y) = e^{i2\pi\alpha(H(y)-1/2)}$, where $H(y)$ is the Heaviside step function, the transverse coordinate $x_T = y$, $\alpha$ is the

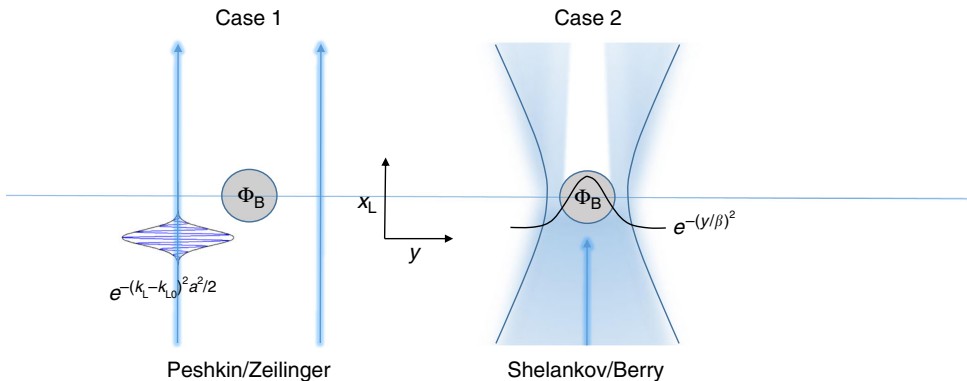

**Fig. 3** Dispersivity theorem and quantum "force". The AB physical system, which involves the passage of electrons (blue) by an area of magnetic flux $\Phi_B$ (gray circle), is analyzed in two ways. In case (1), the effect on a transversely-localized wavepacket with a longitudinal, Gaussian momentum distribution, $e^{-(k_L - k_{L0})^2 a^2/2}$, yields Zeilinger's dispersivity theorem, implying the absence of forces that can lead to time delays. In case (2), the effect on a transverse, Gaussian position distribution yields Shelankov's result to reveal the presence of a quantum "force" that leads to transverse deflection. Note that in both cases the electron wave never penetrates the area of magnetic flux. In physical realizations, the material that supports the magnetic flux area blocks the electron wave

amount of phase shift induced by the interaction, and the transverse momentum, $k_T$, has a Gaussian distribution, $e^{-(k_T - k_{T0})^2 a^2/2}$.

In the momentum representation, Shelankov and Berry show that this can be written as (see also Eq. (1))

$$\varphi_A(k_T, 0) = \left(\frac{\beta^2}{2\pi}\right)^{1/4} e^{-\beta^2 k_T^2/4} \qquad (8)$$
$$\times \{\cos(\alpha\pi) + \sin(\alpha\pi) \mathrm{erfi}(\beta k_T/2)\},$$

where we have chosen $k_{T0} = 0$. Note that this result can be extended to a finite-size fluxtube (Methods). Using the term $F$ to represent the effect of the momentum-dependent interaction, Eq. (8) becomes

$$\varphi_A(k_T, 0) = \left(\frac{\beta^2}{2\pi}\right)^{1/4} e^{-\beta^2 k_T^2/4} F(k_T). \qquad (9)$$

As the complex error function of a real argument is itself real, case 2) can be defined by,

$$F(k_T) = R(k_T) \text{ and } \delta(k_T) = 0. \qquad (10)$$

The expectation value of the position (Eq. (5)) becomes

$$\langle y \rangle = y_0 + \frac{\hbar \langle k_T \rangle}{m} t, \qquad (11)$$

where the expectation value of the momentum needs to be evaluated. To do so, the initial wavepacket, $\varphi_A(k_T, 0) = \left(\frac{\beta^2}{2\pi}\right)^{1/4} e^{-\beta^2 k_T^2/4} R(k_T)$, is used. The momentum term in Eq. (11) is simplified using the antisymmetry of the imaginary error function. The expression used by Berry (Eq. (16) in ref. [28]) can be recovered by identifying $w\theta/\sqrt{2} = \beta k_T/2$, where the deflection angle is given by $\theta \approx k_T/k_{L0}$ and $w$ is a measure of the width of the wavepacket. The final result for the transverse displacement is

$$\langle y \rangle = y_0 + \frac{\hbar}{m\beta} \sqrt{\frac{2}{\pi}} \sin(2\pi\alpha) t, \qquad (12)$$

a non-zero average value that oscillates with the amount of flux enclosed, $\Phi = -\alpha h/e$. In summary, the ZP scenario, and the SB scenario given by Eqs. (6) and (10), respectively, are special cases of the deflection theorem.

**Path integral**. Shelankov's result can be compared to a simulation based on Feynman's path integral approach[36–39]. The path integral and Shelankov approaches are in excellent agreement at the detection plane (Fig. 2) for an initial wavepacket with a transverse phase step

$$\Psi_i(y, 0) = e^{i2\pi\alpha(H(y)-1/2)} e^{-y^2/\beta^2}, \qquad (13)$$

where $y = 0$ is the location of the solenoid, and $\beta$ is the transverse width of the wavepacket. The phase step equals the AB phase, $\varphi_{AB} = -e \int_C \mathbf{A} \cdot d\mathbf{l}/\hbar$, where $\mathbf{A}$ is the vector potential of the magnetic flux, $\Phi$, that is enclosed by the contour, $C$. The phase is independent of distance from the solenoid because $-e \int_C \mathbf{A} \cdot d\mathbf{l}/\hbar = -e\Phi/\hbar = 2\pi\alpha$ for all $C$. The purpose of the path integral simulation is to model the experimental diffraction pattern, where the z-direction for a finite solenoid size (instead of an infinitely thin magnetic flux line) and a shaped aperture (instead of a Gaussian beam) can be taken into account (see Methods). The partial blockage of the electron wave retains the oscillatory deflection predicted by Shelankov and Berry.

**Quantum "force"**. The quantum nature of the force can be understood in the de Broglie-Bohm interpretation of quantum mechanics[37]. The equation of motion for the electron wavepacket can be written as $dp/dt = F_{\text{clas}} + F_{\text{qu}}$ in terms of the classical force, $F_{\text{clas}} = -dV/dy$, and the quantum "force", $F_{\text{qu}} = -dQ/dy$, where the quantum potential is given by $Q = -\hbar^2 \nabla^2 A/2mA$. If the derivative of the quantum potential, $Q$, (with the wavepacket defined as $Ae^{i\phi}$) is non-zero, then there is a local quantum "force". However, this force, which acts on individual de Broglie-Bohm trajectories, is not measurable[40,41]. Operationally, the presence of force is defined by the presence of an average deflection. There is an average deflection if the integral $\int \frac{\partial Q}{\partial y} dy$ is non-zero.

The local derivative $\partial Q/\partial y$ can be calculated for the wavepacket (Eq. (15) in ref. [28]) to be non-zero and finite after the electron has passed the magnetic flux line. The spatial derivative in the quantum potential can not be evaluated immediately after the interaction with the solenoid because the wavefunction is given by a step function. The wavefunction can be propagated for a short distance so it becomes a smooth function. The quantum potential after propagating 1% of the distance from the magnetic flux line to the detection plane is shown in Fig. 4. If the flux line is not magnetized, the quantum potential is left-right symmetric

about $y = 0$. If the flux line is magnetized, the left-right symmetry is broken leading to an average deflection in the far-field diffraction pattern (for $\alpha$ different from 0 and 0.5 modulo 1). The presence of the asymmetric quantum potential supports Keating and Robbins'[29] analysis in terms of a quantum "force" operator. Thus, even in the absence of a classical force (for the AB physical system), a quantum "force" can lead to a non-zero average deflection. Finally, it is interesting to note that an array of flux lines creates a ladder of phase steps, and provides a classical-like force[42]. In a sense this provides a middle ground between a single phase step (quantum "force") and phase slope (classical force).

**Experimental asymmetry observed**. To experimentally verify the predicted probability asymmetry about $y = 0$, we used a transmission

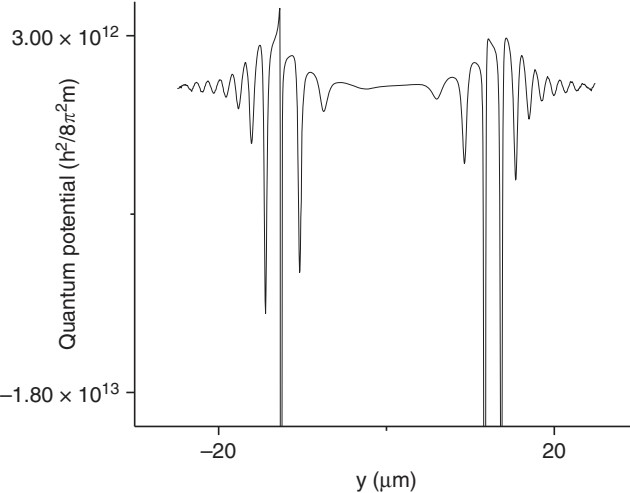

**Fig. 4** The Quantum Potential. The potential, $-\hbar^2 \nabla^2 A / 2mA$, is calculated from the wavepacket $Ae^{i\phi}$ that is propagated 1% of the distance from the magnetic flux line to the detection plane. The wavefunction is obtained from the path integral calculation.The left-right asymmetry is caused by the phase shift induced by the magnetic flux line and illustrates why the word "force" can be used in the present context

electron microscope (TEM) as a versatile electron optical bench tool for quantum experiments. A thin and long ferromagnetic rod was used to create a well-defined magnetic flux line (see Methods). This setup was already successfully applied for mapping specific plasmon modes in nanodevices[43]. A collimated and unfocussed electron beam uniformly illuminates the 5 μm aperture that holds the magnetic nanorod (Fig. 5). The nanorod is 30 μm long, 450 nm wide, 1 μm thick and supports a 65 nm layer of nickel. The 1 μm thickness is sufficient to completely block the electron wave.

A typical far-field intensity profile of the ferromagnetic nickel rod is displayed in Fig. 6, revealing the asymmetric behavior as predicted by Shelankov. It is also in qualitative agreement with the calculations in Fig. 2. In order to demagnetize the nanorod in situ, we exposed the rod to a high intensity electron beam for several hours which led to damage in the nickel film and the loss of its magnetic properties. As the demagnetization occurs in a fairly abrupt fashion, it was not possible to scan through a series of varying magnetizations, and the far-field patterns were recorded only for the fully-magnetized and demagnetized rods. The far-field profile resulting from a demagnetized rod is also displayed in Fig. 6 for comparison, revealing a single symmetric electron diffraction peak as expected. The results of path integral simulations, with magnetic flux line strengths of $\alpha = 0.39$ and $\alpha = -0.02$, show good agreement with the experimental results (Fig. 6, thick, black curves). The counts at each data point are measured with a relative error below 0.005 and are smaller than the data marker size. An inclusive range of $\alpha$ values is provided to illustrate that there is agreement with the expected values of 0.41 (the value experimentally measured by electron holography) and 0.00 (see Fig. 6 caption). The nanorod, which lies in the $y$–$z$ plane, and the diffraction pattern are aligned to within two degrees. In the simulation, the agreement was improved by including partial spatial coherence, which is common in electron microscopy and depends in a sensitive way on the exact setup of the microscope. In particular, the slight positive value in the dip region of the experimental profile is mostly due to partial coherence, with an additional small contribution due to the modulation transfer function of the camera. The average relative y-position of the diffraction pattern with and without magnetization cannot be used to establish the presence of a deflection, as this average position shifts between measurements. The demagnetization and

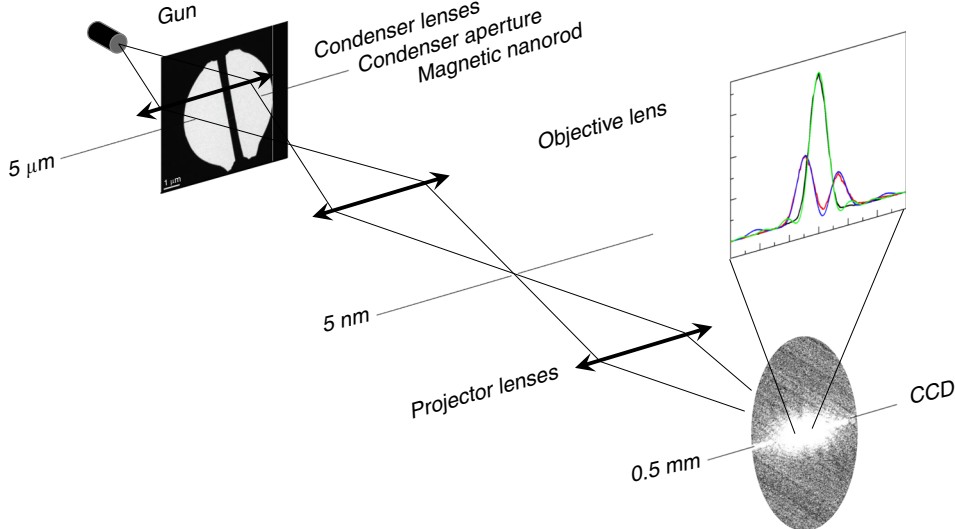

**Fig. 5** Experimental schematic. A magnetized nanorod was placed in an electron microscope in the condenser aperture plane for 60 keV electron energy, and in the sample plane for 300 keV. An electron microscope shadow image is shown. The far-field diffraction pattern was recorded. An example of a raw image is shown

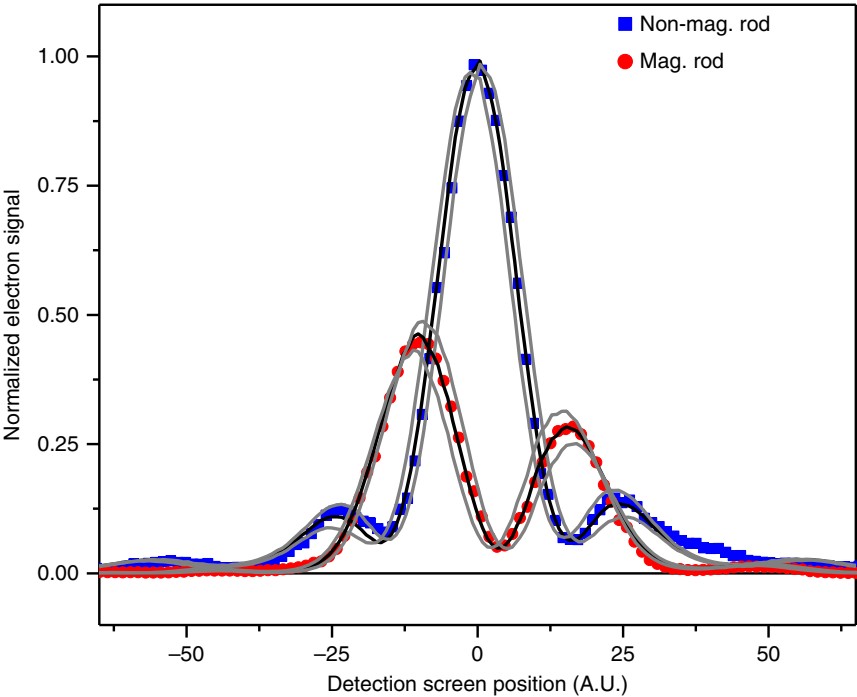

**Fig. 6** Experimental confirmation. An electron diffraction pattern for an aperture that holds a magnetic nanorod (flux line) is measured in the far-field at 60 keV. Magnetized rod experimental data (red dots), path-integral calculation results (thick, black lines, $\alpha = 0.39$, enclosed by thin, gray lines, $\alpha = 0.35 - 0.43$), demagnetized rod experimental data (blue squares) and path-integral calculation results (thick, black lines, $\alpha = -0.02$, enclosed by thin, gray lines, $\alpha = -0.06 - 0.02$) are shown. An overall shift on the screen position is applied for both magnetized and demagnetized experimental diffraction patterns. The results of the path-integral calculations are in agreement with the experimental data and show an asymmetric profile consistent with the predicted spatial deflection, and thus provide indirect evidence of the presence of force

magnetization procedure between the magnetized and non-magnetized measurements and necessary readjustment of the electron beam causes small position shifts in the far-field diffraction pattern. This shift is larger than the predicted deflection, which prevents the direct observation of deflection. Hence, we report only the presence of an asymmetric intensity profile. A future experiment that establishes a non-zero deflection remains highly desirable.

As a verification of symmetry reversal, we placed the nanorod in the image plane of the electron miscroscope where the vicinity of a magnetic lens could be used to flip the magnetization. To that end, the nanorod was rotationally aligned and anti-aligned in situ, and the magnetic field of the lens was ramped up to a high value. The electron energy was set at 300 keV. The accumulated phaseshift is energy-independent and the diffraction pattern for both the 60 keV and the 300 keV data is recorded in the far-field. The result is that the symmetry changes sign with the direction of magnetization (inset Fig. 7).

Additionally the magnetized rod was gradually heated in increments of 10 °C. The resulting reduction in magnetization leads to diffraction symmetry reversals. The last two diffraction pattern reversals and the diffraction above the Curie temperature (when the nanorod is demagnetized) are shown in Fig. 7 together with the path integral simulation. The phase step size for the nanorod was estimated from experimental holographic phase-maps to be $0.58 \pi$ ($\alpha = 0.29$) and $1.32 \pi$ ($\alpha = 0.66$).

In the simulation the position of the diffraction pattern, its height, and the amount of incoherence was fitted.

**Role of fringing fields**. Fringing fields have been considered a confounding factor in AB-type experiments, and could, in principle, lead to distortions of the electron diffraction pattern in our

experiment. The fringing fields for our nanorod have been analyzed in an earlier paper [43]. The finite length of the nanorod causes the presence of fringe fields at the hole through which the electrons pass and determines its strength (the longer the rod, the lower the fields). Thus the nanorod length is by design much longer than the hole diameter, in order to minimise the strength of the fringing field. In these conditions it would be a coincidence if the weak fringing fields at the hole (that emanate from the ends of the nanowire, see Fig. 8a) would yield the asymmetry predicted by Shelankov and Berry. It is interesting to compare our setup to the first experimental report of the AB-effect by Chambers. There were confounding fringing fields, but the experiment nevertheless demonstrated the AB-effect. Similarly, our report is a confirmation of Shelankov's prediction.

A computation of the magnetic field given by a finite continuous solenoid, scaled to the properties of our nanorod (30 μm long, $\alpha = 0.41$) was also performed. We used this field to compute the AB-phase shift for an electron plane wave. The phase variation across half the 5 μm aperture is found to be no more than $0.05\pi$ rad. Approximating this as a constant phase gradient, we estimate the deflection due to magnetic fringing fields: ~5e-8 rad. As the equivalent length of our setup is about 200 m, this would cause a deflection of about 10 μm. Our diffraction pattern's characteristic size, considered to be the distance between the two intensity maxima, is 373 μm. The deflection due to this fringe field is thus relatively small and does not appreciably affect the shape of our diffraction pattern. The shape of the diffraction pattern could be affected by second and higher order phase shifts (for example, a quadratic phase shift), but these effects are much smaller, and thus we can conclude that the shape of the diffraction pattern is dominantly given by the phase step and not by fringing magnetic fields. An experimental

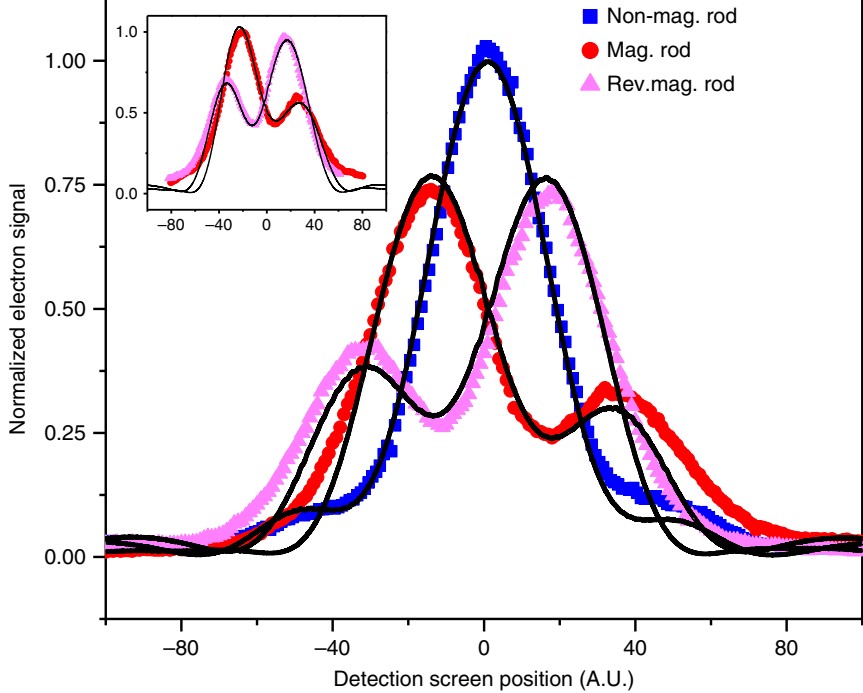

**Fig. 7** Symmetry reversal. Electron diffraction patterns are detected in the far-field at 300 keV for an aperture that holds a nanorod (same rod as in Fig. 6). Three diffraction patterns corresponding to measured phase steps of 1.32 π, 0.58 π, and 0 were recorded. The temperature of the magnetized rod was increased to reach these phase steps. The inset gives magnetization reversal by an external magnetic field

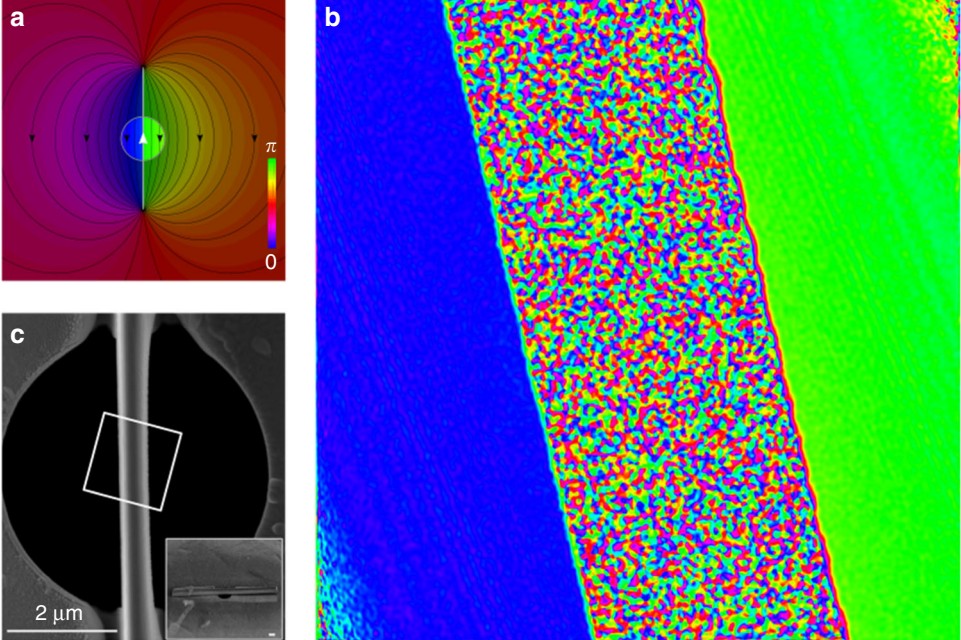

**Fig. 8** Fringe fields. **a** The magnetic field of a 30 μm long magnetized rod is calculated and superimposed over a 5 μm diameter hole. **b** An experimentally measured phase map of the magnetized rod (center multicolored area) and its direct vicinity shows a phase step (blue to green). **c** An electron microscope image of the rod mounted in the middle of the 5 μm diameter hole shows the area (white square) where the phasemap was measured

phase map recorded through electron holography is included in Fig. 8b to illustrate the absence of large phase gradients or large phase distortions. The blue and green regions indicate the wanted phase step while the multi-colored band indicates the magnetized rod where the amplitude is zero and the phase is undetermined. In the lower left corner, some small phase distortion is visible as discussed above. Such high noise areas are due to the fact that it is

impossible to reconstruct the phase in areas where no interference fringes are visible in the original holograms, whether due to strong shadowing from the sample (such as for the 'thick' metal rod) or due to the area being outside the region of interference (top right and bottom left corners). For further relevant examples of magnetic imaging by electron holography see Tonomura[44], Béché et al.[45,46], and Blackburn and Loudon[47]. The region where

the phase map is recorded is overlaid with an electron microscope image of the rod and hole is shown.

## Discussion

In summary, an element of Shelankov's theoretical prediction is confirmed experimentally; a path integral simulation gives good agreement with the experiment and provides the connection between the prediction and experimental result. This gives indirect experimental support for the presence of a quantum "force" in the AB effect.

Theoretically, it is shown that even though Zeilinger's dispersivity theorem is valid for electron propagation in the longitudinal direction, it should not be applied to the transverse direction, and the usual statement that the AB effect is not accompanied by forces is not valid. A theorem is found (Eq. (5)) for which Zeilinger's theorem and Shelankov's result are limiting cases. Classical forces are not needed to explain the observed effect on the electron. Under the assumption that the passing electron does not affect the solenoid, the observed phenomenon remains a pure quantum effect. The observation supports Aharonov and Rohrlich's interpretation that non-local potentials explain the observed phenomenon within a theory that only permits gauge-invariant quantities[48]. The observation does not exclude forces on the nanorod, and thus does not exclude the possibility of Boyer's or Vaidman's descriptions involving force on the flux tube[7,11]. Identification of the momentum terms of the complete system, consisting of the flux tube and electron, including hidden momentum[48,49], may need to be considered in view of the now established presence of a quantum "force"[12].

Even if the experiment detects the presence of a magnetic flux line, it does not offer an approach to search for magnetic monopoles through the detection of Dirac strings (which are themselves examples of magnetic flux lines) as the force is predicted to be zero when the phase shift $2\pi\alpha$ has a value of modulo $\pi$ (Eq. 12). This further highlights the quantum nature of the force. We speculate that the SB force may lead to a new detection mode or architecture for SQUID magnetometry[50], as its counterpart, the longitudinal AB effect, underlies the function of SQUIDs.

## Methods

**Interaction range**. The interaction is assumed to be approximately instantaneous. The purpose of this section is to justify this approximation. The physical system studied is the Aharonov–Bohm one, which for a flux line gives a vector potential that is approximated by $\mathbf{A}(\mathbf{r}) = \frac{1}{2\pi r}\phi_B\hat{\varphi}$, where $\phi_B$ is the magnetic flux carried by the flux line. Integrating over a closed circular particle path that contains the flux line, gives the well-known Aharonov–Bohm phase, $\varphi_{AB} = \frac{e}{\hbar}\oint\mathbf{A}\cdot d\mathbf{l} = \frac{e}{\hbar}\phi_B$. Alternatively, one can integrate over a closed path that consists of two parallel straight-line paths passing on both sides of the flux line and connecting far away from the flux line. The closed loop integral is independent of the loop chosen, which implies that a single straight path phase shift is given by $\frac{e}{\hbar}\int_{-\infty}^{\infty}\mathbf{A}\cdot d\mathbf{l} = \varphi_{AB}/2$. This phase is also independent of distance to the flux line and changes sign for path on the left of right of the flux line. In the near-field diffraction region, defined by $L_{NF} \leq d_{rod}^2/\lambda_{dB}$, where $d_{rod}$ is the nanorod diameter, the single path phase, $\varphi_{NF} = \frac{e}{\hbar}\int_{-L_{NF}}^{L_{NF}}\mathbf{A}\cdot d\mathbf{l}$, is almost complete. For our parameters, $d \approx 500$ nm and $\lambda_{dB} = 2\times 10^{-12}$ m, the near field reaches a distance of about 0.1 m. The accumulated near-field phase shift, $\varphi_{NF}$, equals, for our parameters, $0.99995 \times \varphi_{AB}/2$ for a path passing a hole size dimeter away from the flux line. Paths closer to the flux line have a phase shift closer to $\varphi_{AB}/2$. The effect of this phase gradient $\Delta\varphi/d$ gives an approximate deflection angle of $\theta = \lambda_{dB}\Delta\varphi/2\pi d \approx 2\times 10^{-12}$ rad, where $d = 5$ μm is the hole diameter. This angle is much smaller than the diffraction angle, $\theta_{diff} \approx \lambda_{dB}/d \approx 5\times 10^{-7}$ rad. In the near field, the effect of the phase should not exceed the effect of diffraction from the rod, or, $L_{NF}\theta \leq d_{rod}$, where $\theta = \lambda_{dB}\Delta\varphi/2\pi d$, and $\Delta\varphi \leq \varphi_{AB} \approx \pi$. This condition is also satisfied and motivates the approximation of describing the interaction by a multiplication of the electron wave with an instantaneous phase step at the plane of the flux line.

**Derivation steps of deflection theorem**. To obtain the time-dependent wavepacket in the position-representation, Eq. (4) is transformed to the momentum representation,

$$\begin{aligned}\varphi_F(k,t) &= \frac{1}{\sqrt{2\pi}}\int\psi(x,t)e^{-ikx}dk\\ &= \frac{\sqrt{a}e^{ik_0x_0}}{2\pi^{5/4}}\int\int e^{-(k'-k_0)^2a^2/2}e^{i(-k'x_0-\omega(k')t)}\\ &\quad\times e^{ik'x}F(k')dk'e^{-ikx}dx\\ &= \frac{\sqrt{a}e^{ik_0x_0}}{\pi^{1/4}}e^{-(k-k_0)^2a^2/2}e^{i(-kx_0-\omega(k)t)}F(k).\end{aligned} \tag{14}$$

The expectation value of the position operator $x = i\partial/\partial k$ is evaluated as follows,

$$\begin{aligned}\langle x\rangle &= \int\varphi_F*(k,t)i\frac{\partial}{\partial k}\varphi_F(k,t)dk\\ &= \frac{a}{\sqrt{\pi}}\int\left[e^{-(k-k_0)^2a^2/2}e^{i(-kx_0-\omega(k)t)}F(k)\right]^*\\ &\quad\times\left(i\frac{\partial}{\partial k}\right)\left[e^{-(k-k_0)^2a^2/2}e^{i(-kx_0-\omega(k)t)}F(k)\right]dk\\ &= \frac{a}{\sqrt{\pi}}\int\left[e^{-(k-k_0)^2a^2/2}e^{-i(-kx_0-\omega(k)t)}F*(k)\right]\\ &\quad\times\left[-i(k-k_0)^2a^2+x_0+\frac{\partial\omega}{\partial k}t+i\frac{\partial F/\partial k}{F(k)}\right]\\ &\quad\times e^{-(k-k_0)^2a^2/2}e^{i(-kx_0-\omega(k)t)}F(k)dk.\end{aligned} \tag{15}$$

Setting $F(k) = R(k)e^{i\delta(k)}$, a further simplification is made as follows,

$$\begin{aligned}\langle x\rangle &= x_0 + \frac{a}{\sqrt{\pi}}\frac{\hbar}{m}t\int k|R|^2e^{-(k-k_0)^2a^2}dk\\ &\quad+ \frac{ia}{\sqrt{\pi}}\int\left[(k_0-k)a^2+\frac{1}{R}\frac{\partial R}{\partial k}+\frac{i\partial\delta}{\partial k}\right]|R|^2e^{-(k-k_0)^2a^2}dk\\ &= x_0 + \frac{a}{\sqrt{\pi}}\frac{\hbar}{m}t\int k|R|^2e^{-(k-k_0)^2a^2}dk\\ &\quad+ \frac{ia}{\sqrt{\pi}}\int\left[\frac{\partial}{\partial k}\left(\frac{1}{2}|R|^2e^{-(k-k_0)^2a^2}\right)\right]dk\\ &\quad+ \frac{a}{\sqrt{\pi}}\int\frac{\partial\delta}{\partial k}|R|^2e^{-(k-k_0)^2a^2}dk\\ &= x_0 + \frac{a}{\sqrt{\pi}}\frac{\hbar}{m}t\int k|R|^2e^{-(k-k_0)^2a^2}dk\\ &\quad+ \frac{ia}{\sqrt{\pi}}\int\frac{\partial S}{\partial k}dk+\frac{a}{\sqrt{\pi}}\int\frac{\partial\delta}{\partial k}|R|^2e^{-(k-k_0)^2a^2}dk.\end{aligned} \tag{16}$$

The expectation value of the position operator is simplified to

$$\begin{aligned}\langle x\rangle &= \frac{a}{\sqrt{\pi}}\int e^{-(k-k_0)^2a^2}\left[-i(k-k_0)a^2+x_0\right.\\ &\quad\left.+\frac{\partial\omega}{\partial k}t+i\frac{\partial R/\partial k}{R}\right]|R(k)|^2dk.\end{aligned} \tag{17}$$

Using normalization of the wavepacket and propagation in free space ($\partial\omega/\partial k = \hbar k/m$), it follows that

$$\begin{aligned}\langle x\rangle &= x_0 + \frac{a}{\sqrt{\pi}}\frac{\hbar}{m}t\int k|R|^2e^{-(k-k_0)^2a^2}dk\\ &\quad+ \frac{ia}{\sqrt{\pi}}\int\frac{\partial S}{\partial k}dk+\frac{a}{\sqrt{\pi}}\int\frac{\partial\delta}{\partial k}|R|^2e^{-(k-k_0)^2a^2}dk,\end{aligned} \tag{18}$$

where $S = \frac{1}{2}|R|^2e^{-(k-k_0)^2a^2}$. The term $\int\partial S/\partial k dk$ is zero for functions for which the derivative and the functional value tends to zero at infinity. This is true for all the cases studied here, and implies that high momentum components of the wavepacket are not affected by the interaction. The final result is the deflection theorem expressed in Eq. (5).

**Extension to finite-size flux region**. The phase step result discussed in the "dispersionless and quantum "forces"" section was done for the theoretical construct of a flux line. The analysis can be extended to the case when the magnetic flux provided by the magnetic rod is present in a finite region, or a "flux tube". This is relevant as the experiment is performed for a flux tube. The analysis is done in two ways. The first is an extension of Shelankov's approach in momentum space, the second is by path integration in position space. In the first approach, the starting point is Eq. (3). For the interaction described by a phase step, the wavefunction for a finite magnetic rod size $d$, is given by

$$\begin{aligned}\varphi_d(k_T,0) &= \frac{1}{\sqrt{2\pi}}\int_{-\infty}^{-d/2}e^{-i\alpha\pi}e^{-y^2/2\beta^2}e^{-ik_Ty}dy\\ &\quad+ \frac{1}{\sqrt{2\pi}}\int_{d/2}^{\infty}e^{i\alpha\pi}e^{-y^2/2\beta^2}e^{-ik_Ty}dy\\ &\propto e^{-\beta^2k_T^2/2}\left\{e^{-i\alpha\pi}erf\left(\frac{i\beta k_T}{\sqrt{2}}+\frac{y}{\sqrt{2}\beta}\right)\Big|_{-\infty}^{-d/2}\right.\\ &\quad\left.+e^{-i\alpha\pi}erf\left(\frac{i\beta k_T}{\sqrt{2}}+\frac{y}{\sqrt{2}\beta}\right)\Big|_{-\infty}^{-d/2}\right\}.\end{aligned} \tag{19}$$

In the flux line limit, $d \to 0$, the Shelankov/Berry result (Eq. 8) is recovered. A numerical evaluation of the average deflection with the deflection equation (Eq. 5) using as input the wavefunction Eq. (19) as a function of $d$, is given in Fig. 9 (dashed blue line).

The initial distribution is Gaussian, the same as used by Shelankov and Berry. The path integral for the same initial distribution is given by the solid black line (path integral data points were calculated for 100 nm intervals and connected with straight lines as a guide for the eye). The result is given for $\alpha = 1/4$, when the deflection is largest. (As before, the deflection oscillates with the value of $\alpha$). The agreement between the analytic extension and the path integral result is good. To simulate the experiment an initial tophat distribution (that is uniform over the opening of the aperture) is chosen. The qualitative behavior is the same as for the

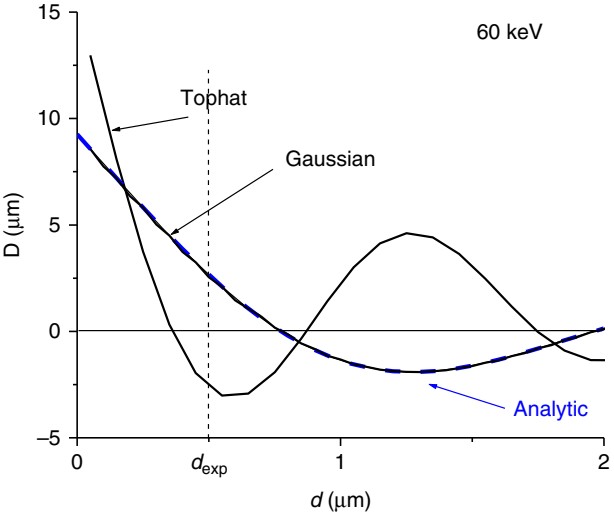

**Fig. 9** Flux tube. The average predicted deflection is given as a function of the magnetic rod size, or in other words, the flux tube diameter $d$. The analytic extension of Shelankov's approach for a flux line to flux tubes is evaluated (blue dashed line). The path integral result (black solid line) is in excellent agreement. The overlapping analytic and path integral calculation results were obtained for an initial Gaussian distribution. For a tophat distribution that corresponds to the experimental initial distribution for an aperture, the path integral result (solid black line) indicates the same qualitative behavior

initial Gaussian distribution. The size of the magnetic rod used in the experiment is labeled with $d_{\text{exp}}$. The typical range over which the deflection becomes small is ~1 μm. The average expectation value was calculated by integrating in the far-field over a 400 μm-wide detection area. This expectation value slowly converges with detection size, while the typical range reduces with the detection size.

To illustrate the magnitude of the effect for $d \to 0$, the deflection (Eq. 16 in ref. [28]) can be written by performing the substitution $w = w_0 k$, where $k = 2\pi/\lambda_{\text{dB}}$ is the wavenumber and $w_0$ is now the waist measured in SI units. Note that Eq. (16) is given in units of deBroglie wavelengths (see bottom page 5628 of ref. [28]). That means that the deflection angle $D$ is of the order $D \cdot 1/w \times \lambda_{\text{dB}}/(2\pi w_0)$. The order of magnitude estimate is done using the following numbers: $w_0 \times 5 \times 10^{-6}$ m, $w \times 5 \times 10^{-6} \times 2\pi/(3 \times 10^{-12}) \times 10^7$, and thus $D \times 10^{-7}$ m. Our electron microscope camera length is about one hundred meters, which gives a deflection on the detection screen of $10^{-5}$ m. Note that this value is smaller than the diffraction pattern pixel size, but of the same order of magnitude.

**Path integral calculation**. In the path integral approach, the final wavepacket $\Psi_f(y, t)$ at the detector plane is given by[36,37],

$$\Psi_f(y,t) = \int N \exp(i\pi l/\lambda_{\text{dB}})\Psi_i(y', 0)\mathrm{d}y', \qquad (20)$$

where the initial wavepacket is $\Psi_i(y', 0)$, $N$ is a normalization factor, and $\lambda_{\text{dB}} = h/p$ is the de Broglie wavelength of the electron. The length of an individual path from some point $y'$ in the interaction plane (parallel to $y$–$z$ located at the solenoid) to $y$ in the detection plane is $l = (D^2 + (y' - y)^2)^{1/2}$, where $D$ is the distance between the interaction and detection planes. Note the absence of a factor of two in Eq. (20) in the phase factor[37] as compared to constructing the matter wave using Huygens' principle by analogy to optics[38,39].

**Sample fabrication and electron microscopy**. A thin film (65 nm) of nickel protected by a gold layer (1 μm) was milled using a focused ion beam (FIB) microscope to obtain a $(30 \times 1 \times 1)$ μm³ ferromagnetic rod. The rod was then deposited with a nano-manipulator over a 5 μm aperture drilled in a SiN grid covered with a 1 μm thick layer of gold (Fig. 4). The rod width was then thinned down to 450 (50) nm. This gave an estimated magnetic flux line strength of α ~ 0.41. The magnetic flux was experimentally assessed using off-axis electron holography. A reference electron wave was superposed with the wave interacting with the flux line. The large aspect ratio between the length and width of the rod allowed for a good approximation of the rod as a single magnetic domain magnetized along its long axis. A detailed description of such an aperture has been given in other work[43]. The aperture with the ferromagnetic rod was inserted in the condenser plane of an FEI Titan³ microscope operating at 60 kV. Objective and projector

lenses were used to image either the magnetic rod or the far-field diffraction plane, as illustrated in Fig. 5. For the 300 kV experiment, the microscope was operated in Lorentz mode (objective lens off). This allowed us to maintain sufficient beam coherence over the aperture. At first, the sample was mounted in a rotation tilt tomography holder, in order to magnetize the magnetic rod in two opposite directions. Indeed, we purposely turned on the magnetic field of the objective lens to 11.5% of its maximal strength (~200 mT) to force the rod magnetization in one or the other direction along the rod axis. The rod was rotated and the holder tilted (+78°) in order to be as parallel as possible to the objective length field, before turning it on. To magnetize the rod in the order direction, the holder was tilted −78° before applying the field. Secondly, the aperture was mounted on a heating holder so that the rod could be heated in situ.

### Data availability
Data available on request from the authors.

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

## Acknowledgements

H.B. would like to thank Michael Berry for bringing the presence of a quantum "force" to our attention. A.B., G.G. and J.V. acknowledge support from the European Research Council under the 7th Framework Program (FP7) ERC Starting Grant 278510 VORTEX. G.G. acknowledges support from the Fonds Wetenschappelijk Onderzoek -Vlaanderen (FWO). M.B. and H.B. acknowledge support by the U.S. National Science Foundation under Grant No. 1602755.

## Author contributions

All authors, M.B., G.G., A.B., J.V., H.B., contributed to all aspects of this work.
