## [Peer Review File · Nature Communications]

Reviewers' comments:

Reviewer #1 (Remarks to the Author):

The paper “Quantum forces and non-dispersivity in the Aharonov-Bohm effect” deals with a quantum effect predicted decades ago but still under active discussion in the literature. The main claim of the paper is the detection of a quantum force acting on the charge in the Aharonov-Bohm conditions, that is, when the classical Lorentz force is absent. As the authors put it, the experimental findings are “...in apparent contradiction to textbook description of the effect”. Besides the experimental part, the paper offers some theoretical advances in understanding of the physical origin of the “quantum force”. No doubt, the claims are novel as such, and the subject area of the paper is of interest for a broad audience,

However, I am not ready to recommend publication of the manuscript in its present form. In my opinion, both, theoretical and experimental, parts of the paper require improvements and more evidence.

1. The author’s theoretical arguments concerning the magnetic force and their general theorem are based solely on Eq.10 which is borrowed from the theory of an ideal Aharonov-Bohm line in Ref.[28]. As clearly stated in [28], Eq.10 gives the scattered-out wave when the centre of the incoming Gaussian beam is aimed exactly at the Aharonov-Bohm line. If the incident beam strikes the flux line non-centrally, Eq.10 is invalid even for a Gaussian wave and the force is exponentially reduced. For a general shape of the incoming beam, supporting the paraxial approximation theory of Ref.[27], numerics in Ref.[28] shows that the magnetic force is absent if the incoming wave vanishes at the position of the flux line: In this case, the outgoing wave remains disturbed by the flux line but without any integral effect, *i.e.*, the beam centre of gravity propagates without deflection to the right or to the left.

In view of these considerations, the generality of the paper theoretical conclusions is questionable. In any case, the authors should address these concerns.

Besides, it is far not clear to what extent a theoretical construction – an ideal flux line – may serve as the model for real life experiments.

2. Citation from Discussion p.7: “A general theorem is found (Eq. 9) for which Zeilinger’s theorem and Shelankov’s result are special cases.”

What is the meaning here? I have honestly tried to comprehend the message but without success. It could be that there is a misprint in the equation number (“Eq.9”), but I am not able to find any other equation that may be considered as “a general theorem”.

What is the content of the general theorem found by the authors? How the theorem is formulated?

3. On p.6, the upper part of the left column: “average y-position... could not be used to establish the presence of a deflection, as the average position shifts between measurements”.

What is the possible source of the shifts? If the average position is not under control, can the deflection be measured in this setup?

4. If the average y-position is not under control, how the curves in Fig. 6 are constructed? Is there any overall shift made by the artist hand for each of the curves?

5. Fig. 6 indeed shows left-right asymmetry in the curve with red circles. The authors ascribe the asymmetry to the rod magnetism.

However, any left-right asymmetric agent like imperfections of the rod shape or the incident beam-rod misalignment, may lead to an asymmetric diffraction picture. A

direct evidence in favour of the magnetic origin of the asymmetry is the sensitivity of the diffraction patterns to the magnetization direction – up or down.

Is it experimentally possible to compare the diffraction patterns observed for the opposite rod magnetization?

If these experiments were performed, what was their result?

6. Compare two statements:

(a) On p.6, the upper part of the left column: “ ...the shift ... prevents the direct observation of deflection”,

(b) Fig.6 Experimental confirmation, The last sentence of the caption: “... calculations are in agreement with the experimental data... and .. consistent with the predicted spatial deflection”.

Are (a) and (b) compatible?

7. I see some are minor problems with the paper.

The text will look better if the elementary transformations in Eqs(5,6, 14) are omitted.

I find myself misled when the authors use words “path integration” and refer to Feynman Ref.[36] in connection with Eq.(M4) and other places in the main text. In my opinion, integration in Eq.(M4) has nothing to do with Feynman’s path integrals, and the authors just exploit Feynman’s authority to justify their procedures. I suggest that instead of Feynman in Ref.[36], they refer to Huygens and/or Fresnel diffraction formula in a book on wave optics (e.g. Born-Wolf).

In conclusion, the paper may become publishable provided the authors finds really convincing arguments to rebut the above concerns.

Reviewer #2 (Remarks to the Author):

The authors present experimental study which sheds light on the controversial topic of the presence of force in the Aharonov-Bohm experiment. In particular, they demonstrate experimentally an old theoretical predictions of Shelankov. It is a challenging experiment and the results are convincing, so it presents significant contribution for the analysis of this subtle issue and I recommend its publication.

The main role of the paper is to present experimental results that will help to analyze the interpretation of the Aharonov-Bohm Effect. It adds discussion of various approaches, but I think that term "force" should be used with more caution. The main result is to reproduce predictions of Shelankov. Shelankov himself used phrases like "as if by the "Lorentz" force" and was careful to write "Lorentz force" in his paper. It has to be made clear that this analysis in which the electron wave function does overlap with the solenoid is not directly connected to the controversies about the cases in which the electron wave does not penetrate the region with the magnetic field. Also it should be made clear that DeBroglie-Bohm force has a different status as it related to motion of Bohmian trajectories. I find this work more related to the discussion of the possible understanding of the AB effect as "as if Lorentz Force" when the electron go through the wall of magnetic field given in Feynman Lecture Notes as nicely discussed in a recent paper <https://link.springer.com/content/pdf/10.1007/s40509-017-0124-z.pdf>

Zeilinger's and Shelankov analyses were done more or less in the same time, so "more recently" in the Abstract sounds strange.

Reviewer #3 (Remarks to the Author):

The authors reports convincing experimental results demonstrating probability asymmetry of the displacement expectation value for electron beams interacting with a magnetized nanorod. This asymmetry value was predicted by Shelankov long ago, and is known to be due to the presence of (quantum forces) which leads to the transversal deflection of the electron waves. Shelankov's theory was in contradiction with the general understanding of the Aharonov-Bohm effect and Zeilinger's theory, which the latter, was based on a one-dimensional scattering theory; hence could deal only with longitudinal forces. These historical steps are very well covered by the authors. In addition to the experiments, the authors show steps in development of a generalized theory, which shows that Shelankov's and Aharonov-Bohm effects are just special cases of this general theory.

The current paper shows very profound scientific descriptions and experimental efforts. The theory is suitably developed for reproducing the experimental results. However, there are some points which needs to be significantly improved to enhance the scientific level of the paper, especially considering the theory.

1st- The whole AB theory is based on the interaction of electrons with magnetic vector potential. Equations (2) to (4) are used to describe such interaction. How should the reader understand the function A ? Is it the magnetic vector potential? I understand these equations as if the whole electron- A interaction is treated as direct multiplication of the electron wave packet with a magnetic vector potential in momentum space. But how and why? The Schrodinger equation regarding the electromagnetic interaction does not imply that, but rather the electromagnetic interaction is described as the multiplication between the electron wave packet and the intensity of the magnetic vector potential in spatial coordination and not in momentum space. In momentum space a convolution might be employed. The authors need to better describe their method from first principle.

2- Again in connection with the first comment, the interaction has been considered instantaneous, without elaborating on the spatial scale of the magnetic vector potential. In my opinion this is an oversimplification.

3- The path integral formalism is indeed only the propagation of the electron in free space, taking the initial wave packet the one which has been suggested considering the assumptions noted above. In my opinion a neat theory should be described here, for example considering the Wolkow states of the electrons including the electromagnetic interaction. How the formalism described in the paper is compared to the Wolkow's analysis?

In addition there are few minor mistakes which should be corrected:

1- Fig. 2 does not have a (b) part as stated in the Figure caption. It is much better to change this figure to color format, encoding each line with a different color.

2- Fig. 5 is a very important result and should be enlarged.

3- In equation (M2), in the 5th row, $-i(k-k_0)^2 a^2$ should be corrected as $-i(k-k_0) a^2$

Reply to Reviewers. Note that the reviewer comments are either repeated below or partially quoted in blue, while our replies are given in black. We hope this facilitates a clear organization. We have taken the reviewers comment very seriously and made significant changes and feel that our resubmitted work is improved significantly, thanks to the helpful comments of the reviewers and editor. Note that changes in the main manuscript text are highlighted as per editor instruction. Also note that figure 3 and 5 have been modified and there is a new figure 7 following reviewer comments.

Reviewer #1 (Remarks to the Author):

Reviewer #1: "...If the incident beam strikes the flux line non-centrally. Eq.10 is invalid even for a Gaussian wave and the force is exponentially reduced. For of a general shape of the incoming beam, supporting the paraxial approximation theory of Ref.[27], numerics in Ref.[28] shows that the magnetic force is absent if the incoming wave vanishes at the position of the flux line: In this case, the outgoing wave remains disturbed by the flux line but without any integral effect, i.e., the beam centre of gravity propagates without deflection to the right or to the left..."

Reply: In a pictorial representation the situation described by the reviewer is shown by figure 1 on the left. What we have in our experiment is described by figure 1 on the right. The probability of the incoming wave is shown superimposed with the finite width fluxline. After passage or during passage the electron wave is completely blocked with the fluxline.

Figure 1.

We have added a sentence below figure 5 to make this clear: "A collimated and unfocused electron beam fully illuminates the 5 μm aperture that holds the magnetic nanorod uniformly." Note that the aperture is illuminated with an electron beam much wider than the aperture diameter. The centering is ensured by the placement of the nanorod on the aperture. Figure 1, 5 and 7 of the paper also convey this information, and Fig. 7 b does this experimentally. Our quantum mechanical simulations shown in Fig. 6 are done for the situation on the right of figure 1 of this reply. If there was a misalignment of the electron beam as indicated in figure 1 left there would be very little difference between the non-magnetized wire calculation and magnetized wire calculation. This is not the case in the experiment nor in the calculation and can thus be ruled out. Finally note the good agreement between experiment and the ab initio theory calculation in figure 6. This supports the assumptions that are made.

We have also added: “The nanorod is 30 μm long, 450 nm wide, 1 μm thick and supports a 10 nm layer of nickel.” We hope clarifies the nanorod geometry.

Reviewer #1: “Citation from Discussion p.7: A general theorem is found (Eq. 9) for which Zeilinger's theorem and Shelankov's result are special cases. What is the meaning here? I have honestly tried to comprehend the message but without success. It could be that there is a misprint in the equation number (Eq.9), but I am not able to find any other equation that may be considered as a general theorem. What is the content of the general theorem found by the authors? How the theorem is formulated?”

Reply: There is indeed a misprint in the equation number. We thank the reviewer for finding this. It should have been Eq. 7 (in the modified manuscript this is now Eq. 5). The “general” theorem was found in the under the section named “General Theorem” (boldface on page 3) that ended with Equation 7. We presume that the referee does not find that the word “general” has been aptly chosen, and we agree. We have renamed the theorem: “Deflection theorem” instead of “General theorem”, and removed the word “general” throughout the paper.

The deflection theorem is derived in the theorem section with intermediate steps shown in the methods section. The content is that if an interaction changes the amplitude and phase of a wavefunction by some general complex function $F(k)$ then we can calculate the average deflection. The theorem is more general than Zeilinger's dispersivity theorem or the Shelankov/Berry result in the sense that both can be obtained from it. In the paper we call this case 1 and case 2 now indicated by Eq. 6 and Eq. 10, the first being a pure phase change as a function of momentum with no amplitude dependence on the momentum k , while the second is a pure amplitude change as a function of momentum with no phase change.

Reviewer #1: “...What it the possible source of the shifts? If the average position is not under control, can the deflection be measured in this setup?”

Reply: The deflection can not be measured in the current version of the setup. In the introduction in paragraph 2 we state this: “A crucial experiment remains necessary to directly demonstrate the electron beam deflection and thus the presence of a quantum force.” The source of the shift is the following. The demagnetization procedure described in the experimental section (for 60 keV) and magnetization reversal procedure (for 300 keV) requires readjustment of the electron beam between measurements. The readjustment causes small position shifts in the far field diffraction pattern. Above figure 6, a sentence has been added to describe this: “The demagnetization and magnetization procedure between the magnetized and non-magnetized measurements and necessary readjustment of the electron beam causes small position shifts in the far field diffraction pattern.”

Reviewer #1: “If the average y-position is not under control, how the curves in Fig. 6 are constructed? Is there any overall shift made by the artist hand for each of the curves?”

Reply: An overall shift is applied for both magnetized and non-magnetized experimental diffraction patterns. We attempted to emphasize this point in our original text with: “This shift is larger than the predicted deflection, which prevents the direct observation of deflection. Hence, we report only the presence of an asymmetric intensity profile.” We have now added to the caption of figure 6: “An overall shift is applied for both magnetized and non-magnetized experimental diffraction patterns.” And we have rewritten the ending sentence of the caption to reemphasize this: “...and thus provide indirect evidence of the presence of force.”

Reviewer #1: “...Is it experimentally possible to compare the diffraction patterns observed for the opposite rod magnetization?...”

Reply: We agree that this is a very important point. This was difficult, but we are happy to report that we have now succeeded in reversing the magnetization and that the resulting observed asymmetry changes sign too! This is now added to the paper, see Figure 7 and the text describing this. To achieve this the nanorod was mounted in a different holder that could be rotated in situ close to a magnetic lens of the electron microscope. Rotationally aligning and anti-aligning the nanorod with the magnetic field, ramping the external magnetic field up to a high value, allowed the magnetization to be flipped. Additionally, the nanorod was heated to reduce the magnetization and demagnetize above the Curie temperature. A minor complication is that the mount is in another observation plane of the microscope, which led to the use of different electron microscope settings including a different electron energy. Note that the accumulated phase shift is energy independent and the diffraction pattern for both the 60 keV and the new 300 keV data is recorded in the far-field. This means that the shape of the pattern is fully characterized.

Reviewer #1: “Compare two statements:

(a) On p.6, the upper part of the left column: “...the shift ... prevents the direct observation of deflection”,

(b) Fig.6 Experimental confirmation, The last sentence of the caption: “... calculations are in agreement with the experimental data... and .. consistent with the predicted spatial deflection”.

Are (a) and (b) compatible?

Reply: They are compatible in a limited sense. The theory predicts that the phase step around the magnetized nanorod does two things: it causes the asymmetry and the deflection. We can observe the asymmetry but not the deflection. The observed asymmetry provides support for the theory and thus indirectly through the theory support for the deflection. We hope that the additions under point 3 and 4 sufficiently clarify this point to the reader. And nowhere in the paper do we intend to imply that we observe the deflection.

Reviewer #1: I see some are minor problems with the paper.

The text will look better if the elementary transformations in Eqs (5,6, 14) are omitted.

Reply: Eqs (5,6) are moved to the methods section, while Eq (14) is removed. The equation numbering has been corrected throughout.

Reviewer #1: I find myself misled when the authors use words “path integration” and refer to Feynman Ref.[36] in connection with Eq.(M4) and other places in the main text. In my opinion, integration in Eq.(M4) has nothing to do with Feynman's path integrals, and the authors just exploit Feynman's authority to justify their procedures. I suggest that instead of Feynman in Ref.[36], they refer to Huygens and/or Fresnel diffraction formula in a book on wave optics (e.g. Born-Wolf).

Reply: Eq. (M4) is equation 23 in Feynman’s original paper. This is not an attempt to exploit Feynman’s authority. Note that the phase factor does not carry the factor $\exp(ikl)$, but $\exp(ikl/2)$. The factor two is specific for Feynman’s approach and not for Huygens see ref [46]. The original Feynman paper was used and translated into computer code used to obtain our results. (The history of this is that David Pritchard’s (MIT) group developed a computer code for matter wave interferometry, Vol. 9, No. 9/September 1992/J. Opt. Soc. Am. A 1601. This was based on Helmholtz’s equation, the optics to matter-optics analogy, and considered to be essentially a Huygens principle, much like the referee suggests. Nevertheless, having worked with Pritchard we became aware of the factor of “two” issue in connecting the time dependent version of matter waves to optical waves as discussed in ref [46] and references therein.) We agree with the referee’s comment that additional references may be useful to the reader. We added Vol. 9, No. 9/September 1992/J. Opt. Soc. Am. A 1601 and a reference to Born-Wolf and the sentence: “Alternatively, one can use the Huygens principle to construct by analogy to optics the matter wave [38,39].”

Reviewer #2 (Remarks to the Author):

Reviewer #2: The authors present experimental study which sheds light on the controversial topic of the presence of force in the Aharonov-Bohm experiment. In particular, they demonstrate experimentally an old theoretical predictions of Shelankov. It is a challenging experiment and the results are convincing, so it presents significant contribution for the analysis of this subtle issue and I recommend its publication.

The main role of the paper is to present experimental results that will help to analyze the interpretation of the Aharonov-Bohm Effect. It adds discussion of various approaches, but I think that term "force" should be used with more caution.

Reply: We agree that caution is required, and that the words to describe this have to be chosen carefully. We hope that the next reply below and the indicated changes made in the paper clarify this issue.

Reviewer #2: The main result is to reproduce predictions of Shelkanov. Shelkanov himself used phrases like "as if by the “Lorentz” force" and was careful to write "Lorentz force" in his paper.

It has to be made clear that this analysis in which the electron wave function does overlap with the solenoid is not directly connected to the controversies about the cases in which the electron wave does not penetrate the region with the magnetic field.

Reply: In this comment, it is implied that in our work the electron penetrates the region with the magnetic field, and that that leads to our work not being directly related to the controversies when electrons do not penetrate, that is the Aharonov-Bohm case, this is false however.

Indeed the electron beam does not penetrate the region of the flux line as electrons are blocked by the 1 μ m thick structure of the nanorod. This is directly evidenced from a TEM image of the aperture (fig. 8) with nanorod which shows negligible amplitude in the nanorod region. To emphasize this we have added to the figure 3 caption: "Note that in both cases the electron wave never penetrates the area of magnetic flux. In a physical realization the material that supports the magnetic flux area blocks the electron wave.". And we have modified figure 3 to illustrate this.

On the issue of the connection with the controversies in the case that the electron wave does not penetrate the region with the magnetic field; first note that, for example, the same happens in Akira Tonomura's famous experiment demonstrating the Aharonov-Bohm effect (Akira Tonomura, Nobuyuki Osakabe, Tsuyoshi Matsuda et al., "Evidence for Aharonov-Bohm Effect with Magnetic Field Completely Shielded from Electron Wave," *Physical Review Letters* 56 (8), 792-795 (1986).) and other experiments on the AB effect. A plane electron wave completely floods the magnetized toroid. Nevertheless the physical structure of the toroid blocks the electron wave and the electron wave only passes in the area of the phasestep (between the open center of the toroid and the outside of the toroid). This is the only part that contributes to the interference pattern. In these areas there is no magnetic field, just as in our case. The physical system is thus directly connected.

The difference between the two cases is that the measured phaseshift for the usual AB-effects experiments (like Tonomura's) are related to the left-right path difference but without being able to look at the overall structure of the interference pattern (this was also not Tonomura's intention). One of the authors HB, has had extensive discussion with Tonomura on the AB-effect (see our joint article in *Physics Today: The Aharonov-Bohm effects: Variations on a subtle theme*, H. Batelaan and A. Tonomura, *Phys. Today* 62 September 38 (2009). Our work on the magnetized nanorod focusses on the overall structure of the interference pattern and considers the same physical system.

The reviewer also points out that the word "Lorentz" force is mentioned in Shelankov's paper. The three papers by Shelankov, Berry, Keating and Robbins, respectively all agree with each other's predictions and that there is no classical Lorentz force (hence the parentheses in Shelankov's paper) and the use of the words "Lorentz force operator" in Keating and Robbins, and Berry writes in his abstract: "...the charged particles... are deflected... vanishing in the classical limit" and in the introduction: "...there is no classical force, there is a quantum force". We have adopted the latter wording to cover the same ideas.

To clarify this issue raised by the referee, we made the following change to the paper: We added a statement and reference on the penetration of the electron wave (in text below Fig.5 caption)

Reviewer #2: Also it should be made clear that DeBroglie-Bohm force has a different status as it related to motion of Bohmian trajectories.

Reply: Yes, we agree. The point made in the paper is that the average deflection, which is an observed quantity and can be calculated with any formulation of quantum mechanics, including quantum deBroglie-Bohm theory, path integral theory, Schrodinger theory, is non-zero. Within the deBroglie-Bohm formulation the nonzero average deflection is due to a nonzero-derivative of the quantum potential (quantum force), while the derivative of the classical potential is zero (classical force). This provides further motivation for the use of the word “quantum force”. As the word “force” has no place in some other formulations of quantum mechanics we believe that this is worth attention (and thus we did our own check). We also believe that we failed to make this sufficiently clear. We have added text (right column page 4) to clarify this point.

Reviewer #2: I find this work more related to the discussion of the possible understanding of the AB effect as "as if Lonretz Force" when the electron go through the wall of magnetic field given in

Feynman Lecture Notes as nicely discussed in a recent paper

<https://link.springer.com/content/pdf/10.1007/s40509-017-0124-z.pdf>

Reply: This is a natural extension of the above comments and we trust that the above replies have clarified this issue.

Reviewer #2: Zeilinger's and Shelankov analyses were done more or less in the same time, so "more recently" in the Abstract sounds strange.

Reply: Agreed. Zeilinger' analyses was first done by Zeilinger and about a decade later by Peshkin. Indeed, the later one by Peshkin was done more or less at the same time as Shelankov's, so we have changed this wording accordingly.

Reviewer #3 (Remarks to the Author):

Reviewer #3: The authors reports convincing experimental results demonstrating probability asymmetry of the displacement expectation value for electron beams interacting with a magnetized nanorod. This asymmetry value was predicted by Shelankov long ago, and is known to be due to the presence of (quantum forces) which leads to the transversal deflection of the electron waves. Shelankov's theory was in contradiction with the general understanding of the Aharanov-Bohm effect and Zeilinger's theory, which the latter, was based on a one-dimensional scattering theory; hence could deal only with longitudinal forces. These historical steps are very well covered by the authors. In addition to the experiments, the authors show steps in

development of a generalized theory, which shows that Shelankov's and Aharonov-Bohm effects are just special cases of this general theory.

The current paper shows very profound scientific descriptions and experimental efforts. The theory is suitably developed for reproducing the experimental results. However, there are some points which need to be significantly improved to enhance the scientific level of the paper, especially considering the theory.

1st- The whole AB theory is based on the interaction of electrons with magnetic vector potential. Equations (2) to (4) are used to describe such interaction. How should the reader understand the function A? Is it the magnetic vector potential?

Reply: "A" is not the vector potential for these equations, but a more general interaction. To avoid confusion we have relabeled the interaction.

Reviewer #3: I understand these equations as if the whole electron-A interaction is treated as direct multiplication of the electron wave packet with a magnetic vector potential in momentum space. But how and why? The Schrodinger equation regarding the electromagnetic interaction does not imply that, but rather the electromagnetic interaction is described as the multiplication between the electron wave packet and the intensity of the magnetic vector potential in spatial coordination and not in momentum space. In momentum space a convolution might be employed. The authors need to better describe their method from first principle.

Reply: Our unfortunate choice of "A" for interaction may have contributed to this comment. The interaction is described as a change of amplitude and phase to allow for a complete change of the wavefunction. In general this can be done in momentum or position space (before a specific interaction is defined). In a later section (page 3 right bottom) we indeed do as the reviewer suggests and transcribe the Shelankov/Berry approach as a phasestep in position space, which they show can be written in momentum representation by our Eq. 8,. We thus agree with the reviewer. We added some description in the theory section to clarify this issue.

Reviewer #3: 2- Again in connection with the first comment, the interaction has been considered instantaneous, without elaborating on the spatial scale of the magnetic vector potential. In my opinion this is an oversimplification.

Reply: This is a simplification. We have now added a section in the methods section to elaborate on the spatial scale of the interaction region, and mention this in the main theory section.

Reviewer #3: 3- The path integral formalism is indeed only the propagation of the electron in free space, taking the initial wave packet the one which has been suggested considering the assumptions noted above. In my opinion a neat theory should be described here, for example

considering the Volkow states of the electrons including the electromagnetic interaction. How the formalism described in the paper is compared to the Volkow's analysis?

Reply: We do not have such a theory available. To our knowledge Volkow states are commonly used for electrons in laser beams (as in for example the Kapitza-Dirac effect) or time dependent electric fields. We are not aware of any treatment of the Aharonov-Bohm physical system in terms of Volkow state, even though we do not rule out that this may be possible. It was not our purpose to find a more rigorous analytical theoretical description. Our purpose was to find a theoretical description that connects the Shelankov/Berry description with the Zeilinger/Peshkin description, and thus we stayed close to the techniques that they used. Additionally, our purpose is to find a simulation that connects the analytic part of theory to a simulation to that allows us to put experimental details such as aperture shape in, to allow comparison between theory and experiment. This was stated in the paper. We acknowledge that our opening statements of the theory part were not sufficiently clear and hope that the rewrite of that section clarifies these issues.

Reviewer #3: In addition there are few minor mistakes which should be corrected:

1- Fig. 2 does not have a (b) part as stated in the Figure caption. It is much better to change this figure to color format, encoding each line with a different color.

Reply: Agree. This has been done and the caption has been corrected.

2- Fig. 5 is a very important result and should be enlarged.

Reply: Agree. This has been done and the caption has been modified.

3- In equation (M2), in the 5th row, $-i(k-k_0)^2 a^2$ should be corrected as $-i(k-k_0) a^2$

Reply: Agree. Thanks for catching this typo. The correction has been made.

Reviewers' comments:

Reviewer #1 (Remarks to the Author):

The authors have improved the paper in many respects. Besides polishing the text, figures and correcting typos, they have added very important new piece of experimental data that show that the asymmetry in the diffraction picture can be flipped by reversal of the magnetization. This observation is most convincing proof that the observed effect is of a magnetic origin. However, I still have some problems with the theoretical part of the paper.

1. The main concern of mine is that the authors extend results developed Shelankov and Berry in [27,28] to the case when there is no guarantee that these results remain valid. Indeed, the Aharonov-Bohm line reconsidered in [27,28] is an ideal construction where the magnetic flux is concentrated in the region of zero size, that is, in the limit where the size is negligibly small compared with the de Broglie wavelength. Obviously, in the experimental conditions when the magnetic rod is of few microns and the electron has the energy of order of 100 keV, this limit is irrelevant. Then it is far not obvious that the interaction can be reduced to a simple phase factor with a step-like change in the phase as in eq. 13. Moreover, in the condition and ideology of the experiment (see Fig.3), "the material that supports the magnetic flux area blocks the electron wave", that is, the rod changes not only the phase but also the absolute value of wave function in the beam. Thus, I doubt eq. 13, if applied to this experiment, and doubt the generality of the "deflection theorem" eq.12 .

2. About deflection. As such, Fig. 1 is a nice and helpful picture illustrating the notion of deflection. However, it seems to be misleading to some extent in the present case. As discussed in [27], the net momentum transfer in the transverse direction $\langle \hbar k \rangle$ is of order of Heisenberg's uncertainty of the transverse momentum in a finite width beam. Therefore, the quantum deflection predicted in [27,28] and described by eq.12 of the manuscripts, should not be viewed in as a rigid shift of the diffracted spot on the observation plane, but as an asymmetric redistribution of the intensity within the spot. The width of the spot grows linearly with time (that is, the distance to the observation plane) and the expectation value follows eq. 12 or its equivalent in [27,28]. Thus, it could be that the authors were too modest and the quantum force could be extracted from their observation of the asymmetric diffraction pattern. The authors may find it interesting to discuss this point.

As a remark, the classical-like deflection as in Fig.1 can be realized in the conditions of the Aharonov-Bohm effect if electrons coherently scattered by an array of the Aharonov-Bohm flux lines (see Shelankov, Phys. Rev. B 62, 3196 (2000)).

3. "Path integration". In their Reply, the authors insist that they indeed apply Feynman's path integration rather than approaches known from optics, and that this is not just words. They insist that in Feynman's approach the matter wave with the wave vector k acquires the phase $\exp(ikl/2)$ (see eq. M6) on the length l rather than $\exp(ikl)$ as numerous previous authors thought. I strongly disagree with this and believe that π in eq. M6 (which in accordance with the Schrodinger-Helmholtz equation describes the Fraunhofer propagation of a matter beam) must be replaced for 2π . On the other hand, the controversial factor of 2 is immaterial in the sense it does not influence any result of the paper. However, it is my duty as a reviewer to express my opinion.

My recommendation to the authors is to make the above replacement or to emphasize this point if they are sure they are right and ready to fight for their belief.

Reviewer #2 (Remarks to the Author):

The authors misunderstood my remark about "Lorentz Force". It is not about Lorentz. It is about " ". I asked to put " " every time "the quantum force" is mentioned. Surely, quantum mechanics is correct. Expectation values of deflection of beam of electrons are correctly calculated and very nicely demonstrated in the paper. But the discussion of forces in the original and the revised version I find misleading. The AB effect (the measured expectation value of the deflection) happens WITHOUT local forces acting on electrons. It can be explained with Bohmian trajectories and introduced Bohmian forces which act on local Bohmian positions. But of course, the law is non-local and these "forces" do not act on physical particles, but on hypothetical hidden variables. Other "forces" are effective average properties. They do not represent any LOCAL forces. The sentence in the discussion: "Momentum conservation of the complete system, consisting of the flux tube and electron, may need to be reconsidered in view of the now newly established presence of a quantum force [12]." is incorrect. Hidden momentum explains all paradoxes.

My comments do not reduce the value of the investigation. Derived equation(s) (such as (5)) are of interest. But it should be made clear that no local force on electron was observed here, with or without putting " " around the word force in the paper.

Reviewer #3 (Remarks to the Author):

The authors have adequately addressed my comments. Although I believe the theory part still can be improved, experiments are convincing and straight forward. I do not have any further comments and suggest publishing of the paper.

Reply to Reviewers. The reviewer comments are repeated below in blue, while our replies are given in black. We hope this facilitates a clear organization. We thank the reviewers for their comments and feel that our resubmitted work is improved significantly, thanks to the helpful comments of the reviewers and editor. Note that changes in the main manuscript text are highlighted as per editor instruction.

Reviewers' comments:

Reviewer #1 (Remarks to the Author):

The authors have improved the paper in many respect. Besides polishing the text, figures and correcting typos, they have added very important new piece of experimental data that show that the asymmetry in the diffraction picture can be flipped by reversal of the magnetization. This observation is most convincing prove that the observed effect is of a magnetic origin. However, I still have some problems with the theoretical part of the paper.

1. The main concern of mine is that the authors extend results developed Shelankov and Berry in [27,28] to the case when there is no guarantee that these results remain valid. Indeed, the Aharonov-Bohm line reconsidered in [27,28] is an ideal construction where the magnetic flux is concentrated in the region of zero size, that is, in the limit where the size is negligibly small compared with the de Broglie wavelength. Obviously, in the experimental conditions when the magnetic rod is of few microns and the electron has the energy of order of 100 keV, this limit is irrelevant. Then it is far not obvious that the interaction can be reduced to a simple phase factor with a step-like change in the phase as in eq. 13. Moreover, in the condition and ideology of the experiment (see Fig.3), "the material that supports the magnetic flux area blocks the electron wave", that is, the rod changes not only the phase but also the absolute value of wave function in the beam. Thus, I doubt eq. 13, if applied to this experiment, and doubt the generality of the "deflection theorem" eq.12 .

Reply: We agree with the reviewer that the finite size of the magnetic rod needs to be addressed and we have now done so in the manuscript's appendix by adding a new section. The reviewer's comment is in agreement with similar comments by Berry [page 5636 of J. Phys. A: Math. Gen. 32 (1999) 5627–5641] : "In any practical experiment, the flux will be contained in a tube of finite radius...On physical grounds, however, I expect that if the flux tube is much thinner than an electron wavelength (a non-trivial practical constraint) the preceding theory can be applied as though the thickness were zero, and there will be a deflection...; this deserves further study." Given this is a main concern of the reviewer, we give here a detailed reply. Note that no claim is made by Berry regarding a fluxtube thickness larger than the electron wavelength, and thus we agree with the reviewer and Berry that we need to address this.

We claim that the deflection remains when the magnetic flux is present in a region of size that is much larger than the deBroglie wavelength λ_{dB} for a finite detector size. The calculation presented in figure 1 in this reply shows this. The figure shows the expectation value of the deflection as a function of the

magnetic rod width, d , while keeping the aperture (through which the electron wave passes) constant at a 5 micron width. The analytical Shelankov fluxline result (the zero size case) is approached in the small d limit. The path integral result (black line) and an analytic extension of Shelankov/Berry result (blue dashed line) are in agreement for an initial Gaussian distribution. For a tophat distribution the result is qualitatively similar. The magnitude of the deflection does not approach zero in a range that is typical of the deBroglie wavelength, but instead in a length scale of the deBroglie wavelength divided by the aperture (or electron wavepacket) width. The fluxline result ($d=0$) is consistent with Equation 16 in Berry's paper J. Phys. A: Math. Gen. 32 (1999) 5627–5641. Equation 16 is given in units of deBroglie wavelengths (see bottom page 5628 of that paper). To illuminate the units, Equation 16 can be rewritten by performing the substitution $w = w_0 k$ where $k = 2\pi / \lambda_{dB}$, is the wavenumber and w_0 is now the waist measured in SI units. That means that the deflection angle D is of the order $D \sim 1/w \sim \lambda_{dB} / (2\pi w_0)$. The order of magnitude estimate is done using the following numbers: $w_0 \sim 5 \times 10^{-6} m$, $w \sim 5 \times 10^{-6} \times 2\pi / (3 \times 10^{-12}) \sim 10^7$, and thus $D \sim 10^{-7} m$. Our electron microscope camera length is about one hundred meters; this gives a deflection on the detection screen of $10^{-5} m$.

Figure 1. Deflection. The main point is that the reduction of deflection with an increasing flux line size d does not occur on a scale of the deBroglie wavelength for a finite detector size.

There is an important qualifier in the above statements: “for a finite detector size.” What happens when the detector size is changed? The result is given in figure 2. For a larger detector the average deflection

Figure 2. Detector size. The main point is that the reduction of deflection with an increasing flux line size d becomes faster for a larger detectors.

decreases faster with increasing magnetic rod width d . Note that for $d=0$ the deflection does not depend on the detector size consistent with Eq. 16. The detector size-dependent comes into play as the Fourier transform gives sidelobes when the center of the Gaussian beam is blocked. The Shelankov/Berry result does not depend on detector size because the Gaussian shape is preserved in the Fourier transform for a fluxline.

Now, we can address the reviewer's final sentence: "Thus, I doubt eq. 13, if applied to this experiment, and doubt the generality of the "deflection theorem" eq.12 ." We agree that we did not address this appropriately in our previous manuscript. We have now added the case of a finite rod size (flux tube) to our Methods section and applied the deflection theorem to it (note that the equation numbering has changed). That is, the wavefunction for the flux tube case is used in our deflection theorem to calculate the deflection; the result is now provided in the section: "Extension to finite size flux regime" in the appendix. These observations are not trivial and we have used the two methods (analytic and path integral) to cross check the results. We have also communicated with Berry for advice. We also attempted a third approach using the analytic solution to Aharonov-Bohm wavefunctions for finite size fluxtubes (Eur. J. Phys. 1 (1980) 154-162), however, we could not make this approach converge. Overall, we think that we took this comment very seriously and agree it deserved attention. We also think that the new appendix section is a significant improvement to our manuscript.

2. About deflection. As such, Fig. 1 is a nice and helpful picture illustrating the notion of deflection. However, it seems to be misleading to some extent in the present case. As discussed in [27], the net momentum transfer in the transverse direction $\langle \hbar k \rangle$ is of order of Heisenberg's uncertainty of the transverse momentum in a finite width beam. Therefore, the quantum deflection predicted in [27,28] and described by eq.12 of the manuscripts, should not be viewed in as a rigid shift of the diffracted spot on the observation plane, but as an asymmetric redistribution of the intensity within the spot. The width of the spot grows linearly with time (that is, the distance to the observation plane) and the expectation value follow eq. 12 or its equivalent in [27,28], Thus, it could be that the authors were too modest and the quantum force could be extracted from their observation of the asymmetric diffraction pattern. The

authors may find it interesting to discuss this point.

We agree that the quantum deflection “should not be viewed in as a rigid shift of the diffracted spot on the observation plane, but as an asymmetric redistribution of the intensity within the spot.” In our experiment, systematic effects cause the “spots” of successive measurements to shift in their position on the detection screen. Thus, we shifted the lineouts of the spots (Figs. 6 & 7 in the paper) using agreement with the theoretical path integral results to determine the amount of shift. A better approach would be to fit the envelopes of the “spots” and then center the envelopes at the center of the detection screen (detection screen position = 0). However, our attempts at fitting the envelope position, and in that way remove systematic errors from the deflection (and other analysis attempts), are inconclusive. So, the correct viewpoint that the reviewer suggests for analysis: “but as an asymmetric redistribution of the intensity within the spot,” has not worked in a rigorous analysis. The systematic error is just a bit too large. We state in the paper that the systematic shift of the center of the diffraction pattern prevents us from measuring the deflection directly. We agree that this is a modest claim given that the asymmetry is linked to the deflection by the theory. And hence the word “indirect evidence.” We would prefer to keep it this way.

As a remark, the classical-like deflection as in Fig.1 can be realized in the conditions of the Aharonov-Bohm effect if electrons coherently scattered by an array of the Aharonov-Bohm flux lines (see Shelankov, Phys. Rev. B 62, 3196 (2000)).

We were not aware of this paper before and have added it to our references. See page 5 and the new reference [42] in the manuscript.

3. "Path integration". In their Reply, the authors insist that they indeed apply Feynman's path integration rather than approaches known from optics, and that this is not just words. They insist that in Feynman's approach the matter wave with the wave vector k acquire the phase $\exp(ikl/2)$ (see eq. M6) on the length l rather than $\exp(ikl)$ as numerous previous authors thought. I strongly disagree with this and believe that π in eq. M6 (which in accordance with the Schrodinger=Helmholtz equation describes the Fraunhofer propagation of a matter beam) must be replaced for 2π . On the other hand, the controversial factor of 2 is immaterial in the sense it does not influence any result of the paper. However, it is my duty as a reviewer to express my opinion.

Please note that the time independent Schrodinger equation is the Helmholtz equation and is associated with the factor of 2π , while the time dependent Schrodinger equation corresponds to Feynman path integral and has the factor of π . Both can be used to solve the problem, but M6 has the time dependence and thus the factor of π .

My recommendation to the authors is to make the above replacement or to emphasize this point if they are sure they are right and ready to fight for their belief.

We follow the recommendation of the reviewer and have now emphasized this point in the paper (page 11), to make sure the reader is aware of this point.

Reviewer #2 (Remarks to the Author):

The authors misunderstood my remark about "Lorentz Force". It is not about Lorentz. It is about " ". I asked to put " " every time "the quantum force" is mentioned. Surely, quantum mechanics is correct. Expectation values of deflection of beam of electrons are correctly calculated and very nicely demonstrated in the paper. But the discussion of forces in the original and the revised version I find misleading. The AB effect (the measured expectation value of the deflection) happens WITHOUT local forces acting on electrons. It can be explained with Bohmian trajectories and introduced Bohmian forces which act on local Bohmian positions. But of course, the law is non-local and these "forces" do not act on physical particles, but on hypothetical hidden variables. Other "forces" are effective average properties. They do not represent any LOCAL forces. The sentence in the discussion: "Momentum conservation of the complete system, consisting of the flux tube and electron, may need to be reconsidered in view of the now newly established presence of a quantum force [12]." is incorrect. Hidden momentum explains all paradoxes.

My comments do not reduce the value of the investigation. Derived equation(s) (such as (5)) are of interest. But it should be made clear that no local force on electron was observed here, with or without putting " " around the word force in the paper.

Reply. We thank the reviewer for the clarification and agree with the statements pertaining to force. We have now added quotes to the term "quantum force".

On the issue of reconsidering momentum conservation, we agree that hidden momentum explains all paradoxes. To do so, the many papers explaining such paradoxes identify the hidden momentum (its time derivative) and all the forces explicitly. Aharonov and Rohrlich concisely state: "The [classical relativistic] paradox is crucial for clarifying the entirely quantum interactions of "fluxons" and charges – the generalized Aharonov-Bohm effect of this chapter", see Chapter 13 opening statement in their book "Quantum Paradoxes" (Wiley, 2004). Given that our manuscript indicates the presence of a "quantum force" that changes momentum and causes deflection, all the terms that affect momentum should be carefully identified. This has to our knowledge not been done. To express this view, we have changed our statement to: "Identification of the momentum terms of the complete system, consisting of the flux tube and electron, including hidden momentum [48], may need to be considered in view of the now established presence of a "quantum force" [12]."

Reviewer #3 (Remarks to the Author):

The authors have adequately addressed my comments. Although I believe the theory part still can be improved, experiments are convincing and straight forward. I do not have any further comments and suggest publishing of the paper.

REVIEWERS' COMMENTS:

Reviewer #1 (Remarks to the Author):

Most of the points raised by the referees have been addressed in the revised version of the manuscript with a new section added to Methods.

As demanded by Referee#2 in the previous round, the authors have put in quotes all the occurrences of 'quantum force'. To my taste, these quotes are not helpful, and their meaning may be a mystery to the reader, especially in the abstract, that is, prior to any explanations. On the other hand, there is no much harm from these quotation marks. As an option, I may suggest the following way to please Referee2: put in quotes only word 'force' in the combination 'quantum force'. Since "force" is rather rare a word in the vocabulary of quantum theory, the quotation marks neither hurt or distract attention of the reader. Of course, this is just an optional suggestion,

I noticed the following minor problems with the text:

1. On p.9 left bottom line there is reference to figure 9. Where is Fig.9? Or the authors mean the figure on p.9 ?
2. The figure on p.9 : What is meant by 'Tophat' and 'Gaussian' in the figure? There are no explanations either in the text or caption.
3. Fig. 3 on p.4, Case 2 . The mathematical expression in the figure should be corrected : the argument of the exponent has wrong dimensionality.

As far as physics of "quantum force" is concerned, I appreciate the caution of the authors when they do not claim they have observed a force in the AB effect. Their only claim is "indirect experimental support for the presence of a quantum force in the AB effect". What they actually observed in their ingenious experiment is an asymmetry in the diffraction pattern induced and controlled by an inaccessible magnetic flux -- indeed, the AB-effect, However, the asymmetry is only a prerequisite for the quantity called "magnetic force" in Ref.[27,28]: the force in Ref.[27,28] is defined as the net transfer of the transverse momentum, that is, an integral property of the diffraction pattern, and the asymmetry as such does not guarantee a nonzero net transfer. For instance, the magnetic force becomes zero if the incoming wave does not overlap with the AB flux line. (Incoming wave should not be confused with the full wave function, which vanishes at the position of the AB-line for any non-integer flux). This feature of the AB-scattering was derived in Ref.[27] in the paraxial approximation and confirmed in Ref.[28] using exact AB-solution. One can show that in the paraxial picture, this result is rather general and the magnetic force (but not the fringes asymmetry!) is absent in any truly AB-configuration, where the magnetic field region is inaccessible for the incoming wave (as in Case2 in Fig.3). However, it remains to be studied if this conclusion survives beyond the paraxial theory; it is also not obvious that the absorption of electrons by the belt around the magnetic rod can be considered just as a partial blockage of the incoming wave. Thus, the current situation is not completely clear, and more theoretical and experiment efforts has to be applied by optimists to prove that a quantum force exists in conditions of the AB-effect.

Although I have some problems with the message of the paper, these doubts are not of the type that prevent me to recommend the manuscript for publication provided the above minor improvements of the text and figures are done.

Andrey Shelankov

Reply to Reviewer. We think we made all the changes requested by the last reviewer. This has clearly improved our manuscript. The changes in the main manuscript text are highlighted in yellow (including those suggested by the editor). Our replies are given below in boldface.

Reviewer #1 (Remarks to the Author):

Most of the points raised by the referees have been addressed in the revised version of the manuscript with a new section added to Methods.

As demanded by Referee#2 in the previous round, the authors have put in quotes all the occurrences of 'quantum force'. To my taste, these quotes are not helpful, and their meaning may be a mystery to the reader, especially in the abstract, that is, prior to any explanations. On the other hand, there is no much harm from these quotation marks. As an option, I may suggest the following way to please Referee2: put in quotes only word 'force' in the combination 'quantum force'. Since "force" is rather rare a word in the vocabulary of quantum theory, the quotation marks neither hurt or distract attention of the reader. Of course, this is just an optional suggestion,

Reply. Agreed. We have followed this suggestion.

I noticed the following minor problems with the text:

1. On p.9 left bottom line there is reference to figure 9. Where is Fig.9? Or the authors mean the figure on p.9 ?

Reply. This has been corrected.

2. The figure on p.9 : What is meant by 'Tophat' and 'Gaussian' in the figure? There are no explanations either in the text or caption.

Reply. The explanation has now been added. (The words referred to the initial electron distribution).

3. Fig. 3 on p.4, Case 2 . The mathematical expression in the figure should be corrected : the argument of the exponent has wrong dimensionality.

Reply. Thanks for catching that! This has now been corrected.

As far as physics of "quantum force" is concerned, I appreciate the caution of the authors when they

do not claim they have observed a force in the AB effect. Their only claim is "indirect experimental support for the presence of a quantum force in the AB effect". What they actually observed in their ingenious experiment is an asymmetry in the diffraction pattern induced and controlled by an inaccessible magnetic flux -- indeed, the AB-effect. However, the asymmetry is only a prerequisite for the quantity called "magnetic force" in Ref.[27,28]: the force in Ref.[27,28] is defined as the net transfer of the transverse momentum, that is, an integral property of the diffraction pattern, and the asymmetry as such does not guarantee a nonzero net transfer. For instance, the magnetic force becomes zero if the incoming wave does not overlap with the AB flux line. (Incoming wave should not be confused with the full wave function, which vanishes at the position of the AB-line for any non-integer flux). This feature of the AB-scattering was derived in Ref.[27] in the paraxial approximation and confirmed in Ref.[28] using exact AB-solution. One can show that in the paraxial picture, this result is rather general and the magnetic force (but not the fringes asymmetry!) is absent in any truly AB-configuration, where the magnetic field region is inaccessible for the incoming wave (as in Case2 in Fig.3). However, it remains to be studied if this conclusion survives beyond the paraxial theory; it is also not obvious that the absorption of electrons by the belt around the magnetic rod can be considered just as a partial blockage of the incoming wave. Thus, the current situation is not completely clear, and more theoretical and experiment efforts has to be applied by optimists to prove that a quantum force exists in conditions of the AB-effect.

Although I have some problems with the message of the paper, these doubts are not of the type that prevent me to recommend the manuscript for publication provided the above minor improvements of the text and figures are done.

Reply. Thank you for the comments, and we agree that more experimental work and theoretical work is needed and hope that this work stimulates such work.